# Yukmijihwang-Tang Suppresses Receptor Activator of Nuclear Factor Kappa-B Ligand (RANKL)-Induced Osteoclast Differentiation and Prevents Ovariectomy (OVX)-Mediated Bone Loss

**DOI:** 10.3390/molecules26247579

**Published:** 2021-12-14

**Authors:** Sang-Yong Han, Yun-Kyung Kim

**Affiliations:** 1Department of Herbal Medicine, College of Pharmacy, Wonkwang University, 460 Iksandae-ro, Iksan 54538, Korea; 030745@daum.net; 2Wonkwang Oriental Medicines Research Institute, Wonkwang University, 460 Iksandae-ro, Iksan 54538, Korea

**Keywords:** Yukmijihwang-tang, osteoclast differentiation, RANKL, NFATc1, OVX, bone loss

## Abstract

Yukmijihwang-tang (YJ) has been used to treat diabetes mellitus, renal disorders, and cognitive impairment in traditional medicine. This study aimed to evaluate the anti-osteoporotic effect of YJ on ovariectomy (OVX)-induced bone loss in a rat and receptor activator of nuclear factor kappa-B ligand (RANKL)-mediated osteoclast differentiation in bone marrow macrophages (BMMs). YJ reduced the formation of tartrate-resistant acid phosphatase (TRAP)-positive multinucleated cells (MNCs) in an osteoclast/osteoblast co-culture system by regulating the ratio of RANKL/osteoprotegerin (OPG) by osteoblasts. Overall, YJ reduced TRAP-positive cell formation and TRAP activity and F-actin ring formation. Analysis of the underlying mechanisms indicated that YJ inhibited the activation of the nuclear factor of activated T cell cytoplasmic 1 (NFATc1) and c-Fos, resulting in the suppression of osteoclast differentiation-related genes such as TRAP, ATPase, H+ transporting, lysosomal 38 kDa, V0 subunit d2, osteoclast-associated receptor, osteoclast-stimulatory transmembrane protein, dendritic cell-specific transmembrane protein, matrix metalloproteinase-9, cathepsin K, and calcitonin receptor. YJ also inhibited the nuclear translocation of NFATc1. Additionally, YJ markedly inhibited RANKL-induced phosphorylation of signaling pathways activated in the early stages of osteoclast differentiation including the p38, JNK, ERK, and NF-κB. Consistent with these in vitro results, the YJ-administered group showed considerably attenuated bone loss in the OVX-mediated rat model. These results provide promising evidence for the potential novel therapeutic application of YJ for bone diseases such as osteoporosis.

## 1. Introduction

Osteoporosis is a common disorder of bone remodeling characterized by loss of bone mass and density, which results in an increased risk of fractures [1]. Osteoporosis is related to various factors, such as age, menopause, and chronic medical conditions, and is correlated with increased mortality in the elderly due to the fact of pathological fractures. It can be classified into two subtypes: type I and type II osteoporosis. Type I osteoporosis, also known as postmenopausal osteoporosis, is caused by estrogen deficiency after menopause, and it mainly affects trabecular bone resorption. Postmenopausal osteoporosis occurs in women, typically between the ages of 50 and 65 years, and signals the end of the fertile phase of the life cycle of a woman. Type II osteoporosis is also called senile osteoporosis and is characterized by age-related loss of cortical and trabecular bone in women and men [2,3,4]. Currently, numerous therapeutic agents are being used to treat osteoporosis. They are usually classified as either anabolic drugs or anti-resorptive drugs. These drugs increase bone mineral density and decrease the risk of skeletal fractures. Anabolic agents, including teriparatide and abaloparatide, increase bone strength and bone formation. Anti-resorptive agents, such as estrogen, bisphosphonates, and selective estrogen receptor modulators (SERMs), increase bone mass by reducing the function of osteoclasts. However, almost all of these drugs have considerable side effects, including osteonecrosis of the jaw (ONJ), hypocalcemia, and gastrointestinal disorders, due to the long-term treatment [5,6]. Therefore, recent experimental studies have focused on the development of new drugs from natural products, including plant extracts, with fewer side effects for the prevention or treatment of osteoporotic bone diseases.

Bone remodeling is a tightly stimulated natural process that requires removal of old bone by osteoclasts and formation of new bone by osteoblasts. Regulation of bone remodeling occurs by various mechanisms related to the interaction between osteoclasts and osteoblasts [7,8]. Osteoclasts are multinucleated giant cells that differentiate from mononuclear cells of the monocyte/macrophage lineage upon stimulation by two fundamental factors: M-CSF and RANKL [9,10]. M-CSF regulates the proliferation and survival of osteoclast precursors via c-Fms and the M-CSF receptor [11]. The TNF superfamily member, RANKL, which is secreted by osteoblasts and activated T cells, is the major factor involved in osteoclast differentiation [12]. OPG acts as a decoy receptor and prevents the interaction of RANKL with its receptor RANK, eventually leading to suppression of RANKL-activated osteoclastogenesis [13]. Thus, the RANKL/RANK/OPG signaling pathway regulates bone remodeling at multiple levels, including osteoclast formation and function [14]. The binding of RANKL to its receptor RANK leads to the recruitment of TNF receptor-associated factors (TRAFs), such as TRAF6, and eventually regulates multiple downstream signaling pathways, including MAPKs, the PI3K/Akt pathway, and NF-κB, leading to the upregulation of the transcription factors c-Fos and NFATc1 that are required for osteoclast differentiation [15,16,17]. Subsequently, activated NFATc1 induces the expression of several osteoclast-specific genes, such as OSCAR, TRAP, calcitonin receptor, cathepsin K, MMP-9, Atp6v0d2, DC-STAMP, and OC-STAMP, which are important for osteoclast formation and function [18,19,20,21,22]. YJ is a multi-herb formula that is widely used in East Asia. YJ consists of six herbal medicines including Rehmanniae Radix Preparata, Corni Fructus, Poria, Dioscoreae Rhizoma, Alismatis Rhizoma, and Moutan Cortex Radicis. Traditionally, YJ has been used to treat clinical kidney yin deficiency symptoms such as hectic fever, dizziness, dry mouth and throat, excessive thirst and liquid intake, spermatorrhea, night sweats, red tongue with less coating, and rapid pulse [23]. Several pharmacological properties of YJ have been reported such as protection against renal ischemia/reperfusion [24], improvement in learning and memory [25], anti-obesity [26], anti-diabetic [27], and antioxidant effects [28]. In particular, YJ repressed RANKL-induced osteoclast differentiation in RAW264.7 cells [29] and reduced parathyroid hormone-mediated bone loss and bone resorption in OVX rats [30]. In this study, we examined the pharmacological effects of YJ on RANKL-mediated osteoclast differentiation in BMMs and animal models of OVX-induced bone destruction.

## 2. Results

### 2.1. High-Performance Liquid Chromatography (HPLC) Analysis of YJ Water Extracts

HPLC analysis simultaneously identified five marker components of YJ: 5-hydroxymethyl-2-furaldehyde (5-HMF) of Rehmanniae Radix Preparata, Alisol A of Alismatis Rhizoma, diosgenin of Dioscoreae Rhizoma, loganin of Corni Fructus, and paeonol of Moutan Cotex Radicis. The five components were recorded with the same retention times as window as studied previously. Alisol A (1), 5-HMF (2), diosgenin (3), loganin (4), and paeonol (5) (Figure 1).

### 2.2. YJ Inhibited Osteoclast Differentiation in the Co-Culture System of BMCs and Osteoblasts

We examined whether YJ was concerned in the formation of TRAP-positive MNCs in the co-culture system of bone marrow cells (BMCs) and osteoblasts. As shown in Figure 1a, YJ markedly inhibited prostaglandin E2 (PGE2)- and 1,25-dihydroxyvitamin (vit D_3_)-activated formation of TRAP-positive MNCs in the co-culture system. YJ significantly reduced the number of PGE2-and vitamin D_3_-mediated TRAP-positive osteoclasts in a dose-dependent manner (Figure 2b,c). Cell cytotoxicity was not affected by treatment with YJ (Figure 2d). We next examined the effects of YJ on the expression of RANKL and OPG in co-cultured cells treated with PGE2 and vit D_3_ using reverse transcription polymerase chain reaction (RT-PCR). Treatment of co-cultured cells with PGE2 and vit D_3_ enhanced the expression of RANKL and suppressed the expression of OPG. YJ had a suppressive effect on the upregulation of RANKL mRNA expression but had promotive effects on the OPG mRNA expression levels (Figure 2e). To confirm these results, the expression levels of RANKL and OPG mRNA were assessed by quantitative real-time PCR. As shown in Figure 2f, the mRNA expression of RANKL was significantly lower, while that of OPG was consistently higher when the cells were treated with YJ compared with the PGE2- and vit D_3_-treated control. These results demonstrate that YJ negatively regulated the formation of osteoclasts in the co-culture system by modulating the expression of RANKL and OPG.

### 2.3. YJ Inhibited RANKL-Mediated Osteoclast Differentiation in BMMs

To verify the effects of YJ in RANKL-mediated osteoclast differentiation, primary BMMs were treated with M-CSF and RANKL in the presence or absence of 25, 50, 100, or 200 μg/mL YJ. In the YJ-untreated group, mature TRAP-positive MNCs occurred. However, YJ treatment reduced osteoclast differentiation as shown in images from a light microscope (Figure 3a). YJ treatment significantly inhibited the number of TRAP-positive MNCs (Figure 3b). These inhibitory effects of YJ did not affect the cell viability at any of the concentrations (25, 50, 100, and 200 μg/mL) (Figure 3c). In addition, formation of the F-actin ring by fusion of osteoclasts was suppressed by YJ (Figure 3d). In this study, TRAP-positive MNCs were defined as having three or more nuclei. As shown Figure 3, we observed that the TRAP-positive multinucleated cells (MNCs) were indicated at three concentration of YJ (25, 50, and 100 μg/mL). On the basis of these results, we were concerned about the possibility of the group treated with 25, 50, and 100 μg/mL of YJ (the number of TRAP MNCs at each concentration were 145, 63, and 18 on average) differentiating into osteoclasts later, because in the figure, the cells treated with 25, 50, and 100 μg/mL of YJ still had multiple nuclei. Therefore, we used 200 μg/mL of YJ that did not show the possibility of osteoclast differentiation.

### 2.4. YJ Inhibited RANKL-Mediated MAPK Signaling Pathway in BMMs

To examine the mechanism underlying the YJ-mediated suppression of osteoclast differentiation, we examined the effects of YJ on RANKL-induced early activation of MAPKs (including p38, JNK, and ERK), Akt, and NF-κB signaling pathways, which are fundamental for osteoclast differentiation and function. RANKL-induced phosphorylation of p38, JNK, ERK, and Akt reached the maximum level at the time point of 5 min, whereas it markedly decreased in the group treated with 200 μg/mL of YJ (Figure 4a). In addition, we evaluated the effect of YJ on the phosphorylation of NF-κB and degradation of IκB by Western blotting. RANKL stimulation led to the phosphorylation of NF-κB and degradation of IκB, whereas YJ treatment decreased RANKL-induced NF-κB and IκB expression at 5 min (Figure 4b). Furthermore, we investigated the inhibitory effect of YJ on NF-κB transcriptional activity in 293T cells using the luciferase reporter assay. YJ significantly reduced the NF-κB transcription activity in a dose-dependent manner (Figure 4c). These results indicate that YJ might abrogate osteoclastogenesis via the inhibition of RANKL-mediated activation of p38, JNK, ERK, Akt, and NF-κB.

### 2.5. YJ Inhibited RANKL-Mediated NFATc1 and c-Fos Expression in BMMs

NF-kB is known to be the regulated upstream factor of NFATc1 and c-Fos during RANKL-activated osteoclastogenesis [31]. To identify the inhibitory effect of YJ on osteoclast differentiation, we measured the RANKL-induced activation of NFATc1 and c-Fos. c-Fos mRNA and protein expression levels reached a maximum at 6 h after exposure to RANKL, while those of NFATc1 were elevated after 24 h of RANKL incubation and peaked at 48 h. Quantitative real-time RT-PCR analysis indicated that YJ significantly suppressed c-Fos and NFATc1 expression at 6 and 48 h, respectively (Figure 5a,b). At the same time, RANKL-mediated mRNA expression of c-Fos and NFATc1 was lowered by YJ, in a dose-dependent manner (Figure 5c,d). In addition, Western blot analysis showed that NFATc1 and c-Fos protein expression were prominently decreased by YJ treatment (Figure 5e,f). NFATc1 is a transcription factor for the last stages of osteoclast differentiation and is activated by RANKL stimulation via nuclear translocation. To further determine whether YJ reduced NFATc1 translocation in the nucleus, we investigated the effect of YJ on the nuclear localization in mature osteoclasts. RANKL-treated cells demonstrated augmented nuclear translocation of NFATc1 compared to the controls. However, YJ stimulation attenuated the RANKL-induced translocation of NFATc1 to the nuclei (Figure 6a). We also measured the nuclear translocation of NFATc1 using nuclear/cytosol fraction kit. RANKL significantly promoted the nuclear translocation of NFATc1; however, it was blocked by YJ treatment (Figure 6b). These results demonstrated that the suppression of NFATc1 and c-Fos activation and blockage of nuclear translocation of NFATc1 are critical steps in the reduction of osteoclast differentiation by YJ.

### 2.6. YJ Reduced RANKL-Mediated Expression of Osteoclast-Related Genes in BMMs

To elucidate the mechanisms underlying the inhibitory effect of YJ on RANKL-activated osteoclast differentiation, we determined the expression of osteoclast-related genes. Our data showed that the mRNA expression of TRAP, Atp6v0d2, OSCAR, OC-STAMP, DC-STAMP, MMP-9, calcitonin receptor, and cathepsin K were significantly upregulated upon exposure to RANKL at the time point of 48 h (Figure 7). However, YJ downregulated the mRNA expressions of TRAP and OSCAR, which were correlated with osteoclast differentiation. Moreover, YJ significantly attenuated mRNA expressions of Atp6v0d2, OC-STAMP, and DC-STAMP, which are required for osteoclast fusion (Figure 7). The expressions of transcription factors correlated with bone resorption, such as calcitonin receptor, MMP-9, and cathepsin K, were markedly reduced by YJ treatment (Figure 7). These results further showed that the inhibitory effect of YJ on RANKL-induced osteoclast differentiation, resorption, and fusion could be mediated through regulation of the activation of osteoclast-related genes.

### 2.7. YJ Led to Increased Bone Density in the OVX Rat Model

Based on the anti-osteoclast differentiation effect of YJ in vitro, the in vivo anti-osteoporotic effect of YJ was tested in an experimental rat model of OVX-induced bone loss. 3D visualization of the femoral area by micro-CT revealed a loss of trabecular bone by OVX, which was reduced in the OVX- and YJ-treated rat (Figure 8a). Morphometric analysis of femurs of OVX rats indicated a decrease in bone volume/tissue volume (BV/TV), trabecular thickness (Tb.Th), and trabecular number (Tb.N) in comparison to the sham group. The BV/TV and Tb.N levels were considerably restored by 200 and 400 mg/kg YJ treatment; however, Tb.Th remained unaffected in these rats (Figure 8b). In contrast, trabecular separation (Tb.Sp) was higher in the OVX group than that in the sham group and decreased as a result of YJ treatment (Figure 8b). Furthermore, histological analysis revealed that YJ treatment reduced OVX-induced bone loss in the femurs of OVX rats (Figure 8c). These findings demonstrated that YJ restores OVX-mediated bone destruction.

## 3. Discussion

In the present study, we described that YJ inhibited RANKL-activated osteoclast differentiation and suppressed F-actin ring formation. YJ also showed an inhibitory effect on osteoclastogenesis in a co-culture of osteoblastic cells and bone marrow cells. Furthermore, YJ downregulated RANKL-induced activation of MAP kinases (p38, ERK, and JNK), Akt, IκB, and NF-κB, and the subsequent expression of NFATc1 and c-Fos. In addition, YJ reduced RANKL-mediated NFATc1 nuclear translocation. Consecutively, it inhibited RANKL-induced activation of osteoclast-related genes including TRAP, Atp6V0d2, OSCAR, OC-STAMP, DC-STAMP, calcitonin receptor, MMP-9, and cathepsin K. This study attempted to assess the inhibitory effects of YJ on osteoclast differentiation in vitro. All in vitro experiments were conducted with a BMM-derived osteoclast culture system.

In 2011, Shim et al. [29] reported four marker components of YJ extracts that were identified using high-performance liquid chromatography (HPLC) analysis. The marker components included paeoniflorin and paeonol in Moutan Cortex Radicis, loganin in Corni Fructus, and 5-hydroxymethyl-2-furaldehyde in Rehmanniae Radix Preparata. In addition, these results further confirmed that YJ inhibits osteoclast differentiation though inhibiting RANKL-activated MAPKs, NF-κB activation, and transcription factors in RAW264.7 cells [29]. Recently, various studies reported that paeoniflorin simultaneously suppresses osteoclastogenesis and facilitates osteoblastogenesis by manipulating the actions of NF-κB, whereas paeonol inhibits RANKL-mediated osteoclastogenesis by suppressing p38, ERK, and NF-κB pathways [32,33]. Moreover, 5-HMF inhibits adipogenesis and enhances osteoblastic differentiation of mesenchymal stem cells (MSCs) derived from rat bone [34]. In this study, we identified 5-HMF, Alisol A, diosgenin, loganin, and paeonol from YJ water extract using HPLC analysis. Based on these results, it was demonstrated that the synergistic and complementary effects of these marker components of YJ may contribute to the inhibitory effect of YJ on RANKL-mediated osteoclast differentiation.

Considering the inhibitory effects of YJ on RANKL-mediated osteoclast differentiation and function in vitro, we evaluated the restorative effects of YJ in OVX-mediated bone loss model in vivo. In this study, YJ improved the reduced BV/TV, Tb.Th, and Tb.N levels in OVX-induced rats in a dose-dependent manner (200 and 400 mg/kg/day), whereas the increment of Tb.Sp was totally reversed. Ha et al. reported that a 13 week repeated oral administration of aqueous YJ extract in SD rats indicated no toxicity and the no-observed-adverse-effect-level (NOAEL) was found to be 2000 mg/kg/day [35]. Additionally, 300 mg/kg YJ promoted longitudinal bone growth by stimulating chondrocyte proliferation [36]. Based on these results, we used a maximum 400 mg/kg YJ, once a day, for oral administration in this study.

In conclusion, this study demonstrates that YJ diminished RANKL-activated osteoclast differentiation via signaling pathways, including MAPKs, Akt, IκB, NF-κB, and NFATc1, and osteoclast differentiation-related gene expression in vitro. Moreover, YJ restores bone density in OVX-induced bone destruction in vivo. These findings may help understand the molecular mechanisms of YJ and provide therapeutic opportunity for the treatment of osteoclast-associated bone diseases. 

## 4. Materials and Methods

### 4.1. Animals and Reagents

The animal experiments were conducted in accordance with the guidelines for animal experimentation by the Institutional Committee of Wonkwang University (Approval number: WKU16-52). All animals were obtained from Samtako Bio Inc. (Osan, Korea). The animals were housed under conditions of controlled temperature of 22 ± 1 °C with a 12 h light and 12 h dark cycle. Human RANKL and M-CSF were purchased from PeproTech EC Ltd. (London, UK). PGE_2_ and vit D_3_ were purchased from Sigma-Aldrich (St. Louis, MO, USA). ERK, phospho-ERK, JNK, phospho-JNK, p38, phospho-p38, Akt, phospho-Akt, IκB, phospho-IκB, NF-κB, and phospho-NF-κB were obtained from Cell Signaling Technology Inc. (Beverly, MA, USA). NFATc1, c-Fos, lamin B, and β-actin were acquired from Santa Cruz Biotechnology, Inc. (Santa Cruz, CA, USA).

### 4.2. Preparation of YJ

The YJ formula was purchased from Omniherb Corporation (Daegu, Korea). The composition of YJ was as follows: Rehmanniae Radix Preparata (16 g), Corni Fructus (8 g), Dioscoreae Rhizoma (8 g), Poria (6 g), Alismatis Rhizoma (6 g), and Moutan Cortex Radicis (6 g). All herbs were immersed in distilled water for 1 h and then extracted by boiling at 100 °C for 2 h. The boiled extracts were percolated through filter paper, and the filtrate was concentrated using a rotary evaporator. The concentrate was lyophilized by a freeze dryer (Bondiro, Ilshin, Korea) and stored at −20 °C. The lyophilized powder was re-suspended in distilled water and used as YJ (Yield: 25.15%).

### 4.3. HPLC Analysis

The Agilent 1200 series HPLC system (Agilent Technologies, Santa Clara, CA, USA) consisted of a quaternary pump VL (G7111A), autosampler (G7129A), ICC column oven heater (G7129A), and variable wavelength detector (G7114A). Analyses were carried out on an ZORBOX Eclipse Plus C18 column (4.6 × 250 mm, 5 μm) at a flow rate of 0.4 mL/min. The detection wavelength was set at 260 nm. The injection volume was 5 μL, and the column temperature was maintained at 30 °C. The mobile phase consisted of the solvent A (10% acetonitrile in water containing 0.1% formic acid) and solvent B (water) (Table 1).

### 4.4. Cell Culture and Osteoclast Differentiation

BMCs were separated by marrow flushing of femurs and tibiae excised from male ICR mice (5 weeks old) and cultured in α-MEM supplemented with 10% FBS and 1% antibiotics. BMCs were seeded onto 100 mm dishes in medium supplemented with M-CSF (10 ng/mL) for 1 day. After 1 day, non-adherent cells were collected and cultured in the presence of M-CSF (30 ng/mL). After 3 days of incubation, adherent BMMs on dish bottoms were considered as osteoclast precursors and used for subsequent experiments. For osteoclast differentiation, BMMs were cultured in α-MEM medium with M-CSF (30 ng/mL) and RANKL (100 ng/mL) for 4 days. Following this, cells were fixed in 3.7% formalin, permeabilized with 0.1% Triton X-100, and stained in TRAP solution. TRAP-positive multinucleated cells (MNCs) with more than three nuclei were counted as osteoclasts.

### 4.5. Co-Culture of Osteoblastic and BMCs

Primary osteoblastic cells were obtained from calvariae of neonatal ICR mice (1 day old). Briefly, after isolation, calvariae were digested with 0.2% dispase (Roche, Mannheim, Germany) and 0.1% collagenase (Sigma–Aldrich, St. Louis, MO, USA) for 5 min. Remaining tissues were sequentially digested 4 times for 10 min with the medium obtained after each digestion. Supernatants were accumulated and used as primary osteoblasts. For the co-culture experiments, osteoblasts (1.5 × 10^4^ cells/100 mm dish) and BMCs (3.5 × 10^4^/100 mm dish) were co-cultured in medium with 10 nM vit D_3_ and 100 nM PGE_2_ for 7 days.

### 4.6. F-Actin Ring Staining

BMMs were incubated for 4 days with M-CSF and RANKL in the presence or absence of YJ. When the osteoclasts were formed, the cells were washed and fixed with 4% paraformaldehyde for 20 min and permeabilized using 0.1% Triton X-100 for 15 min. Cells were blocked with 2% BSA for 1 h and then stained with phalloidin (Molecular Probes, Eugene, OR, USA) for 1 h. Nuclei were stained with DAPI (0.1 µg/mL) for 1 min. The images were acquired using a fluorescence microscope (EVOS FL, AMG, Westburg, Leusden, The Netherlands).

### 4.7. Cytotoxicity Assay

Cytotoxicity was monitored by Cell Proliferation kit II (XTT) (Roche, Mannheim, Germany) according to the supplier’s recommendations. BMMs (1 × 10^4^ cells/well, 96-well plate) were cultured in α-MEM medium supplemented with M-CSF (30 ng/mL) for 24 h. Cells were treated with YJ (25, 50, 100, or 200 μg/mL). After 3 days, XTT solution (50 μL) was added to each well and then incubated for 4 h at 37 °C. Absorbance was measured at 450 nm using a microplate reader (Molecular Devices, Sunnyvale, CA, USA).

### 4.8. Quantitative Real-Time RT-PCR Analysis and RT-PCR

Total RNA was isolated using the Isol-RNA lysis reagent (PRIME, Gaithersburg, USA) according to the manufacturer’s instruction. The RNA samples (1 pg) were reverse-transcribed into cDNA using the ReverTra Ace qPCR RT Kit (Toyobo, Osaka, Japan) according to the supplier’s recommendations. The quantitative real-time RT-PCR was performed using a SYBR^®^ Green Realtime PCR Master Mix (Toyobo, Osaka, Japan) and Step-One Plus™ RT-PCR system (Applied Biosystems, Foster City, CA, USA). Target gene expression levels were normalized to the expression of the endogenous GAPDH gene and calculated from the cycle threshold (Ct) value using the 2^−∆∆^Ct method. Primers used for quantitative real-time RT-PCR are summarized in Table 2. For RT-PCR analysis, the PCR products were electrophoresed on a 1.5% agarose gel and visualized by ethidium bromide staining.

### 4.9. Western Blotting 

Whole cells were washed with 1 × PBS and then lysed with ice-cold lysis buffer (Biosesang, Seongnam, Korea) consisting of Na_3_VO_4_ and protease inhibitor cocktail. Cell lysates were centrifuged for 20 min, and the supernatants were collected. Total proteins were resolved by 10% sodium dodecyl sulfate-polyacrylamide gel electrophoresis and transferred to polyvinylidene difluoride membrane (Bio-Rad, Hercules, CA, USA). The membranes were blocked with 5% BSA for 1 h and incubated with primary antibodies overnight. These antibodies were probed using HRP-conjugated secondary antibodies. Chemiluminescent signals were detected with Western ECL solution (Bio-rad, Hercules, CA, USA). Nuclear/cytosol proteins were isolated using a nuclear/cytosol fractionation kit (BioVision, Mountain View, CA, USA).

### 4.10. Luciferase Reporter Assay

293T cells were cultured in DMEM medium containing 10% FBS and plated for 24 h before transfection in a 96-well plate at 2 × 10^4^ cells/well. The cells were co-transfected with TRAF6 and NF-κB luciferase reporter vector (pGL3-Basic Vector; Promega, Madison, WI, USA) for 3 h in serum-free medium. After 12 h, the transfected cells were incubated with or without YJ. The cells were lysed with 1× lysis buffer at room temperature for 15 min, and luciferase activity was assayed using a luciferase assay system (Promega, Madison, WI, USA).

### 4.11. Immunocytochemistry Analysis

BMMs were stimulated with M-CSF and RANKL in the presence or absence of YJ. After 48 h, the cells were fixed in 4% paraformaldehyde (Tech & Innovation, Gangwon, Korea) and permeabilized with 0.1% Triton X-100. After blocking with 5% BSA for 1 h, the cells were incubated with anti-NFATc1 antibody (1:500). Cells were rinsed with 1 × PBS and incubated with the secondary antibody (1:200) for 1 h and stained with DAPI. Fluorescent images were examined using a fluorescence microscope.

### 4.12. OVX-Mediated Bone Loss Model

Female Sprague–Dawley (SD, 8 weeks old) rats were randomly divided into five groups. After 1 week acclimatization, rats were anesthetized with a ketamine–xylazine mixture (5:1, I.P.), and the ovaries were removed bilaterally. One week after surgery, rats were divided into five main experimental groups (6 rats/group): (1) sham-operated (SH); (2) OVX; (3) OVX rats which received 17β-estradiol (E2, 100 µg/kg once daily, S.C.); (4) OVX rats treated with low doses of YJ (200 mg/kg, P.O.); (5) OVX rats treated with high doses of YJ (400 mg/kg, P.O.). Osteoporosis induced by ovariectomy is associated with estrogen deficiency; 17β-estradiol was used as the positive drug. The rats were euthanized at 12 weeks by diethyl ether inhalation, and the femurs were isolated for micro-CT (Skyscan 1076, Bruker, Kontich, Belgium) and histological analysis.

### 4.13. Micro-CT and Histological Analysis

The left femur was examined with a Skyscan 1076 scanner at a 35 μm resolution. The X-ray source was set at a 50 kV accelerating voltage and 100 μA beam current with a 0.5 mm aluminum (Al) filter. Raw scan images were reconstructed by Skyscan NRecon software (ver.1.6.10.1, Bruker, Kontich, Belgium). Reconstructed images were segmented to allow for trabecular bone structure quantification using CTAn software (ver.1.18.4.0, Bruker, Kontich, Belgium). 3D image visualization was performed with the Ant software (ver.2.4, Bruker, Kontich, Belgium). The right femurs were fixed in 4% paraformaldehyde for 1 day, decalcified in 12% ethylenediaminetetraacetic acid (EDTA, pH 7.4) for 3 weeks at room temperature. Then, femurs were embedded in paraffin. Serial sections were sliced into 5 μm using a microtome (RM2125, Leica Microsystems, Bannockburn, IL, USA) and were subjected to H&E staining.

### 4.14. Statistical Analysis

The results were expressed as the mean ± standard deviation. Statistical analyses were performed by the Student’s *t*-test and one-way ANOVA using SPSS Software (Korean version 12.0; SPSS Inc., Chicago, IL, USA). The Student’s *t*-test was used for comparisons of two means, and one-way ANOVA followed by post hoc analysis using the LSD test was carried out for multiple comparisons. To check for normal distribution and homogeneity of variance we used Levene’s test. The Levene’s test statistical significance was greater than 0.05, assuming equal variances. In the case of unequal-variances, we performed the Kruskal–Wallis test and the Mann–Whitney U test, which are nonparametric statistics. Values of *p* < 0.05 were considered to represent a significant difference.

## Figures and Tables

**Figure 1 molecules-26-07579-f001:**
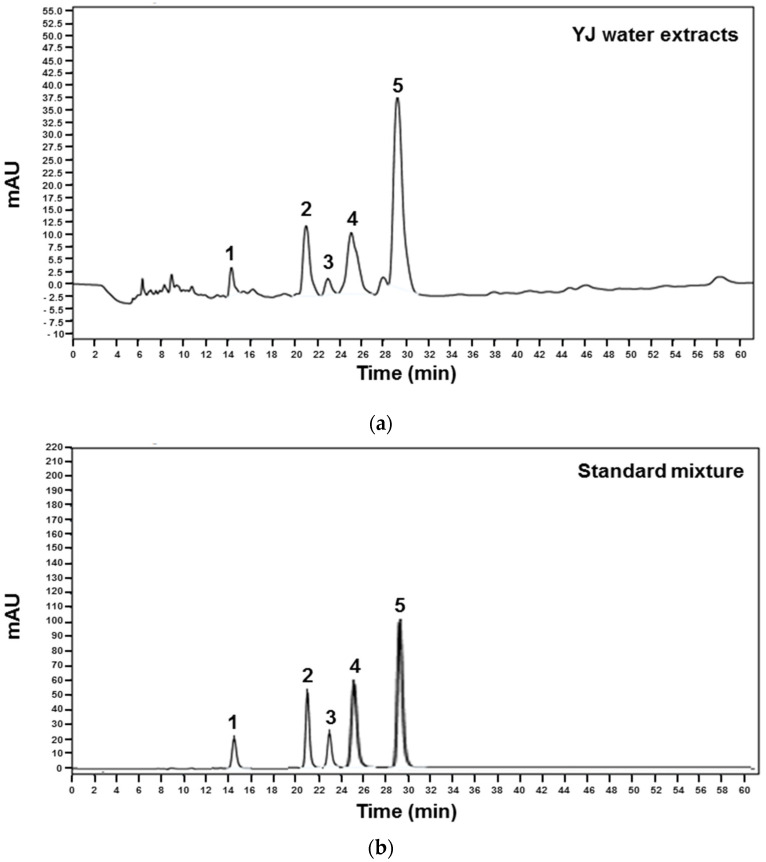
HPLC analysis of the YJ water extracts (**a**) and its reference standard (**b**). Peak identification: 1—Alisol A; 2—5-hydroxymethyl-2-furaldehyde (5-HMF); 3—diosgenin; 4—loganin; 5—paeonol.

**Figure 2 molecules-26-07579-f002:**
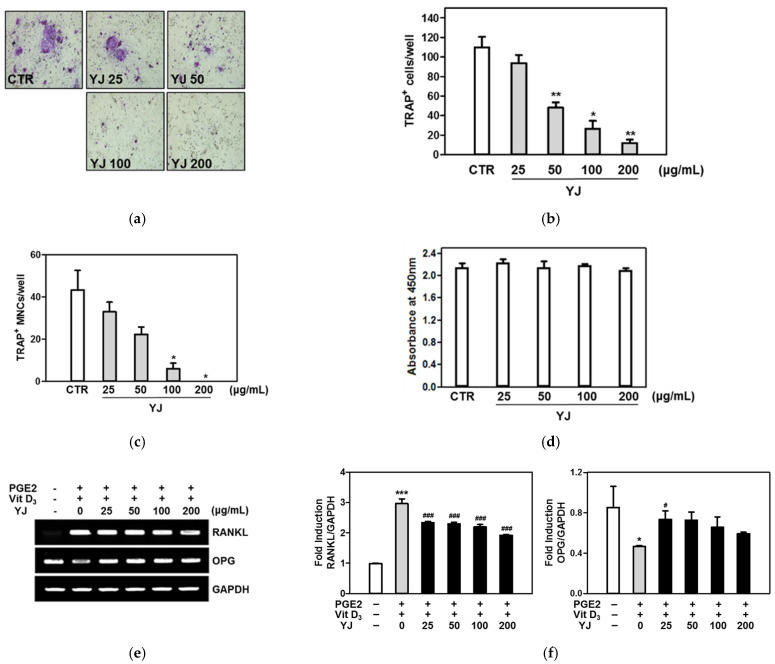
YJ suppressed osteoclastogenesis in the co-culture of bone marrow cells (BMCs) and mouse calvaria-derived osteoblasts. (**a**) BMCs and mouse calvaria-derived osteoblasts were co-cultured for 7 days in a medium containing vit D_3_ and PGE2 in the presence of YJ. Cells were fixed, permeabilized, and stained with TRAP solution. Images were captured under a light microscope (100× magnification). (**b**,**c**) TRAP-positive cells and TRAP-positive MNCs with more than three nuclei were counted. (**d**) Co-cultured cells were seeded into a 96-well plate and cultured for 4 days. After 4 days, cell viability was analyzed using the XTT assay. (**e**) Total RNA was extracted from the cells, and the mRNA levels of OPG, RANKL, and GAPDH (glyceraldehyde 3-phosphate dehydrogenase) were measured using RT-PCR. For RT-PCR analysis, the PCR products were electrophoresed on a 1.5% agarose gel and were visualized by ethidium bromide staining. (**f**) Quantitative real-time RT-PCR was performed to determine the mRNA expression of OPG and RANKL. Data are presented as the mean ± SD of four independent experiments. * *p* < 0.05, ** *p* < 0.01 *** *p* < 0.001 vs. the control group and ^#^ *p* < 0.05, ^###^ *p* < 0.001 vs. the vitamin D_3_- and PGE2-treated group.

**Figure 3 molecules-26-07579-f003:**
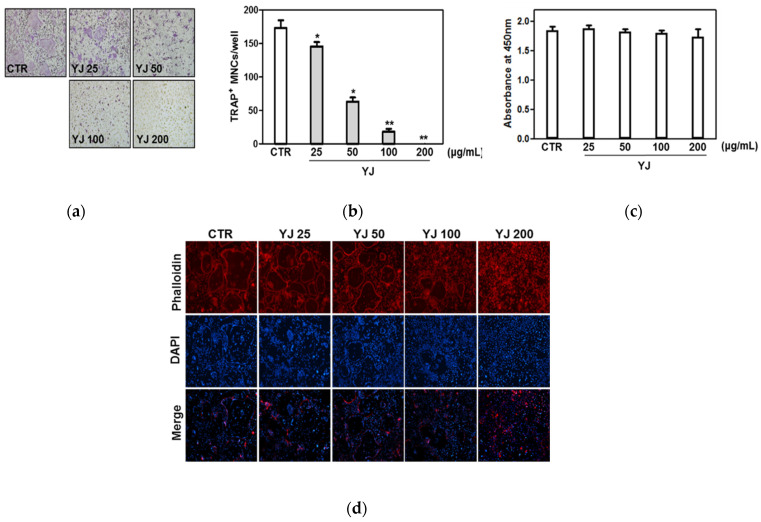
YJ suppressed RANKL-mediated TRAP-positive osteoclast formation. Bone marrow macrophages (BMMs) were cultured in differentiation medium containing M-CSF (30 ng/mL) and RANKL (100 ng/mL) for 4 days with vehicle or YJ (*n* = 4). (**a**) BMMs were fixed, permeabilized, and stained with TRAP solution. Images were captured under a light microscope (100× magnification). (**b**) TRAP-positive MNCs were counted on day 4. TRAP-positive MNCs with >3 nuclei were counted as osteoclasts. (**c**) After YJ treatment for 4 days, cell viability was measured by XTT assay. (**d**) Cells were fixed with 3.7% formalin, permeabilized with 0.1% Triton X-100, and stained with phalloidin (red) and DAPI (blue). Images were obtained by fluorescence microscope (100× magnification). The data presented are the mean ± SD of four independent experiments. * *p* < 0.05, ** *p* < 0.01 vs. the control group.

**Figure 4 molecules-26-07579-f004:**
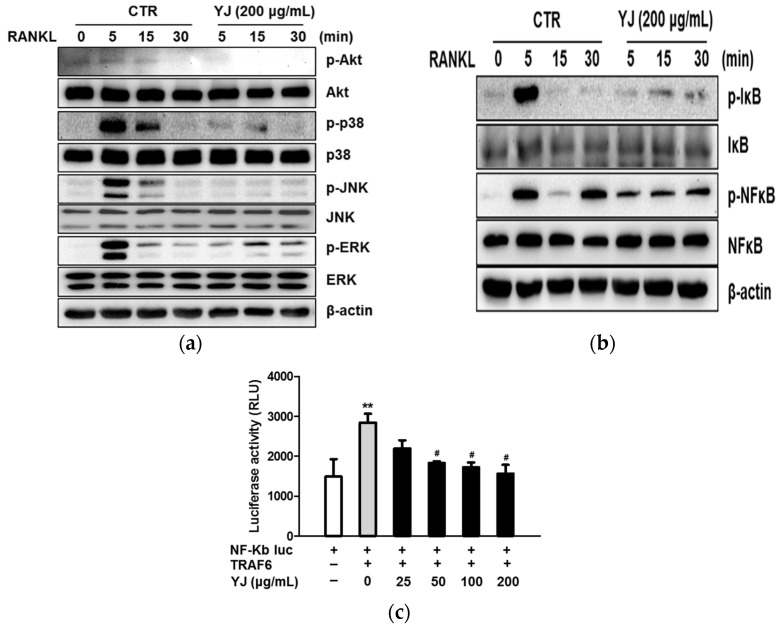
YJ suppresses RANKL-mediated early signaling in osteoclast differentiation. (**a**,**b**) BMMs were serum-starved for 2 h and pre-treated with YJ (200 μg/mL) for 1 h and then treated with RANKL (100 ng/mL) for the indicated time. The cell lysates were evaluated by Western blotting. (**c**) 293T cells were co-transfected with TRAF6 and NF-κB-luciferase reporter plasmid. Luciferase activity was assayed using a luciferase assay system. The data presented are the mean ± SD of three independent experiments. ** *p* < 0.01 vs. the control group and ^#^ *p* < 0.01 vs. the TRAF6-stimulated group. TRAF6, TNF receptor-associated factor 6.

**Figure 5 molecules-26-07579-f005:**
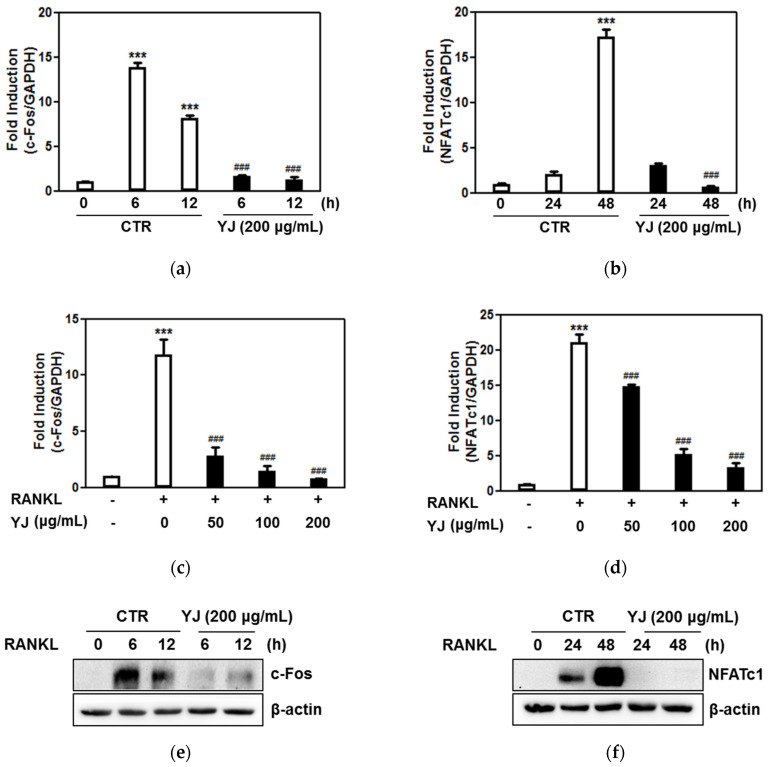
YJ suppressed RANKL-mediated expression of c-Fos and NFATc1 in BMMs. BMMs were treated with M-CSF and RANKL in the presence or absence of YJ (200 μg/mL) for the indicated time. (**a**–**d**) Total RNA was isolated from cells and mRNA expression levels of c-Fos and NFATc1 were measured by quantitative real-time RT-PCR. (**e**,**f**) The cell lysates were subjected to Western blot analysis with c-Fos and NFATc1 antibodies. β-actin was used as a loading control. The data are presented as the mean ± SD of three independent experiments. *** *p* < 0.001 vs. the control group and ^###^ *p* < 0.001 vs. the RANKL-treated group at indicated times and concentrations.

**Figure 6 molecules-26-07579-f006:**
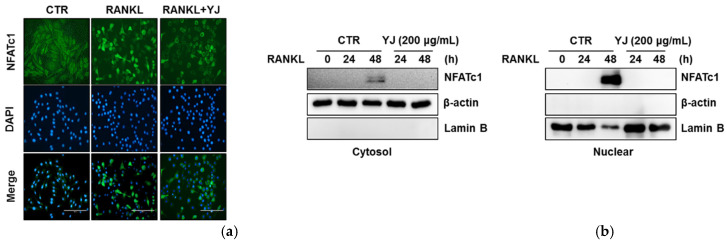
YJ suppresses RANKL-mediated nuclear translocation of NFATc1. The BMMs were treated with M-CSF and RANKL in the presence or absence of YJ (200 μg/mL). (**a**) Cells were fixed with 4% paraformalin, permeabilized with 0.1% Triton X-100, and stained with NFATc1 antibody (green) and DAPI (blue). (**b**) Nuclear and cytoplasmic fractions were analyzed using Western blot analysis. Lamin B and β-actin served as the loading controls for nuclear and cytoplasmic proteins, respectively. The data are presented as the mean ± SD of three independent experiments.

**Figure 7 molecules-26-07579-f007:**
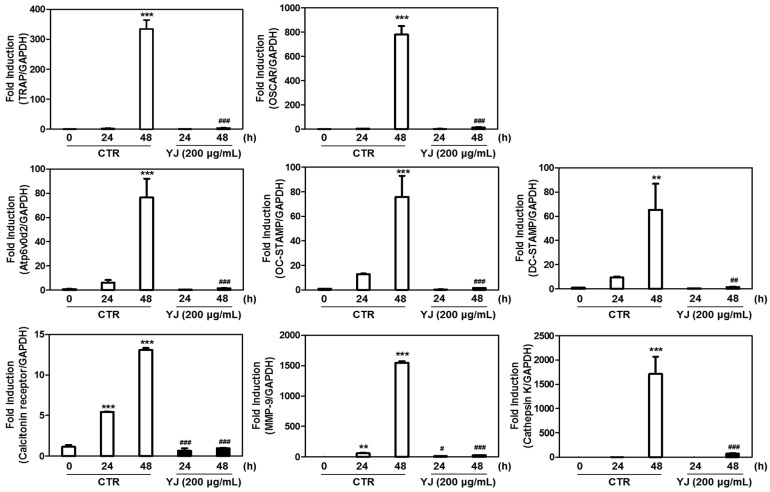
YJ suppresses RANKL-mediated expression of osteoclast-related genes in BMMs. BMMs were treated with M-CSF and RANKL in the presence or absence of YJ (200 μg/mL). Total RNA was isolated from cells and the mRNA expression of the indicated gene was analyzed by quantitative real-time RT-PCR. The data are presented as the mean ± SD of three independent experiments. ** *p* < 0.01, *** *p* < 0.001 versus the control at 0 h; ^#^ *p* < 0.05, ^##^ *p* < 0.01, ^###^ *p* < 0.001 versus the control at each time point. TRAP, tartrate resistant acid phosphatase; OSCAR, osteoclast-associated receptor; Atp6v0d2, vacuolar-type H^+^-ATPase V0 subunit D2; OC-STAMP, osteoclast stimulatory transmembrane protein; DC-STAMP, dendritic cell-specific transmembrane protein; MMP-9, matrix metalloproteinase-9.

**Figure 8 molecules-26-07579-f008:**
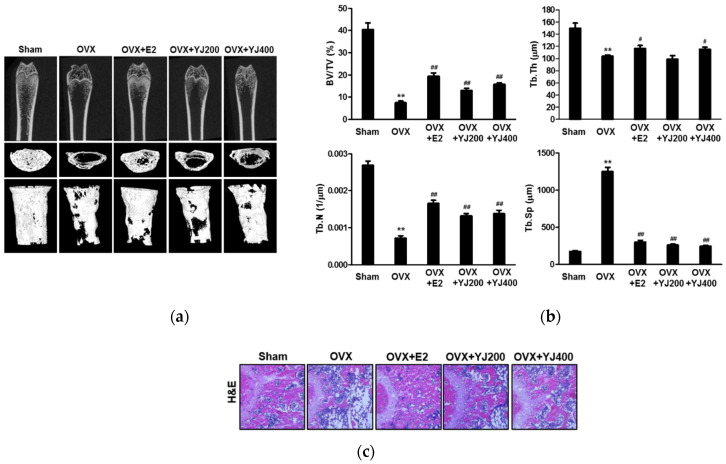
YJ prevented OVX-mediated bone loss. Nine-week-old SD rats were ovariectomized, and one week later, YJ (200 and 400 mg/kg) or PBS was administered orally once a day for 12 weeks. All rats were sacrificed with diethyl ether inhalation at 12 weeks. (**a**) Representative 3D reconstruction images of femurs from micro-CT. (**b**) The BV/TV, Tb.Sp, Tb.Th, and Tb.N were quantified at the left femur. (**c**) Dissected femora were fixed, decalcified, embedded in paraffin, and sectioned. Sections were stained with hematoxylin and eosin (H&E). ** *p* < 0.01 versus SH group; ^#^ *p* < 0.05, ^##^ *p* < 0.01 versus the OVX group (*n* = 6).

**Table 1 molecules-26-07579-t001:** Mobile phase condition of chromatographic separation.

Time (min)	A	B	Flow Rate (mL/min)
0	15	85	0.4
30	40	60	0.4
40	60	40	0.4
60	100	0	0.4

**Table 2 molecules-26-07579-t002:** Sequences of real-time RT-PCR primers.

Target Gene	Primer Sequence (5′–3′)	
*c-Fos*	Forward	CTGGTGCAGCCCACTCTGGTC
	Reverse	CTTTCAGCAGATTGGCAATCTC
*NFATc1*	Forward	CAACGCCCTGACCACCGATAG
	Reverse	GGCTGCCTTCCGTCTCATAGT
*TRAP*	Forward	ACTTCCCCAGCCCTTACTAC
	Reverse	TCAGCACATAGCCCACACCG
*OSCAR*	Forward	GAACACCAGAGGCTATGACT
	Reverse	CCGTGGAGCTGAGGAAAAGG
*Atp6v0d2*	Forward	TCAGATCTCTTCAAGGCTGTGCTG
	Reverse	GTGCCAAATGAGTTCAGAGTGATG
*OC-STAMP*	Forward	TCACTGACCTGCGTTTCGACAA
	Reverse	GCGTAGGCCTGTAGCCACCAA
*DC-STAMP*	Forward	GCAAGGACCCCAAGGAGTCG
	Reverse	CAGTTGGCCCAGAAAGAGGG
*Cathepsin K*	Forward	ACGGAGGCATTGACTCTGAAGATG
	Reverse	GTTGTTCTTATTCCGAGCCAAGAG
*Calcitonin receptor*	Forward	TGGTTGAGGTTGTGCCCA
	Reverse	CTCGTGGGTTTGCCTCATC
*MMP-9*	Forward	TCCAACCTCACGGACACCC
	Reverse	AGCAAAGCCGGCCGTAGA
*GAPDH*	Forward	ACCACAGTCCATGCCATCAC
	Reverse	TCCACCACCCTGTTGCTGTA

## Data Availability

The data presented in this study are available within this article.

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
