# Peer review of "Yukmijihwang-Tang Suppresses Receptor Activator of Nuclear Factor Kappa-B Ligand (RANKL)-Induced Osteoclast Differentiation and Prevents Ovariectomy (OVX)-Mediated Bone Loss"

_molecules, 2021, doi:10.3390/molecules26247579_

Round 1

Reviewer 1 Report

The author should provide clear figure 1, and explain why has two spectra in this figure 1. 

Author Response

We appreciate your comment. Figure 1 has been corrected. Related information were described in Figure 1 legend.

Reviewer 2 Report

  1. The authors should include their response on a Comment #12 into the manuscript body.

Author Response

We appreciate your comment. According to your comment, we added Comment #12 to section 2.3

This manuscript is a resubmission of an earlier submission. The following is a list of the peer review reports and author responses from that submission.

Round 1

Reviewer 1 Report

To ensure the quality of Yukmijhwang-tang (YJ), the HPLC or LC-MS profile of YJ should be given in this manuscript. And then label the main compounds in this profile. 

Reviewer 2 Report

The manuscript submitted by Han and Kim is not yet ready for being reviewed. In order for a reviewer to critically evaluate the results of experimental studies and the statistical evaluation thereof, it is essential to have information on the number of biological and technical replicates. In other words, n=?. This seems not to be mentioned anywhere.

Furthermore, the authors state that ALL statistical analyses were done using only one-way ANOVA - a parametric test. However, it is obvious that the authors have used post-tests, but it is not stated which tests were used. Also e.g. Figure 6 clearly shows data that are grouped - thus one-way ANOVA cannot possibly have been used - so what then?

Also the authors do not state anywhere how they tested for normality of their data-set, allowing them to use a parametric test such as one-way ANOVA.

So without these very elemental information it is impossible to review this study.

Reviewer 3 Report

The paper describes the effects of YJ on bone system in mice. It was shown that YJ suppresses osteoclast differentiation and bone loss.

Comments

  1. Title: all abbreviations should be disclosed.
  2. Throughout the text: All the abbreviations should be disclosed the first time they appear.
  3. The language of the paper should be improved.
  4. Lines 71-75: It not clear, to which cells in the co-culture related RNA expression data. This should be clarified.
  5. Figures 1-6: Control groups should be indicated in each graph.
  6. Figure legends 1-6: The authors should indicate, which cells exactly they describe in each study.
  7. Discussion: Lines 202-233 are related to Introduction and should be moved to Introduction.
  8. Lines 217-219: This sentence is incorrect. It should be repharased.
  9. Line 236: The abbreviations should be disclosed.
  10. Line 294: The authors should indicate which cell proliferation kit was used.
  11. Lines 300-308: The authors should describe how quantification of relative gene expression was performed.
  12. General comment: It is not clear why the authors conducted their study using the highest concentration of YJ (200ug/ml) if the effect was observed at concentration of 25-50ug/ml? The authors should present additional set of data for all the experiments using YJ at concentration of 25 or 50ug/ml.